# Integrative analysis of genomic variants reveals new associations of candidate haploinsufficient genes with congenital heart disease

Enrique Audain[1,2°], Anna Wilsdon[3°], Jeroen Breckpot[4], Jose M. G. Izarzugaza[5], Tomas W. Fitzgerald[6], Anne-Karin Kahlert[1,2,7], Alejandro Sifrim[8,9], Florian Wünnemann[10], Yasset Perez-Riverol[11], Hashim Abdul-Khaliq[12], Mads Bak[13,14], Anne S. Bassett[15,16], Woodrow D. Benson[17], Felix Berger[18], Ingo Daehnert[19], Koenraad Devriendt[4], Sven Dittrich[20], Piers EF Daubeney[21], Vidu Garg[22,23,24,25], Karl Hackmann[7], Kirstin Hoff[1,2], Philipp Hofmann[1,2], Gregor Dombrowsky[1,2], Thomas Pickardt[26], Ulrike Bauer[26], Bernard D. Keavney[27,28], Sabine Klaassen[29,30,31], Hans-Heiner Kramer[1,2], Christian R. Marshall[32,33], Dianna M. Milewicz[34], Scott Lemaire[35], Joseph S. Coselli[36], Michael E. Mitchell[36], Aoy Tomita-Mitchell[36], Siddharth K. Prakash[34], Karl Stamm[36], Alexandre F. R. Stewart[37], Candice K. Silversides[15], Reiner Siebert[38,39], Brigitte Stiller[40], Jill A. Rosenfeld[17], Inga Vater[39], Alex V. Postma[41,42], Almuth Caliebe[39], J. David Brook[3], Gregor Andelfinger[43], Matthew E. Hurles[44], Bernard Thienpont[4,45], Lars Allan Larsen[13]*, Marc-Phillip Hitz[1,2,39,44]*

1 Department of Congenital Heart Disease and Pediatric Cardiology, University Hospital of Schleswig-Holstein, Kiel, Germany, 2 German Center for Cardiovascular Research (DZHK), Kiel, Germany, 3 School of Life Sciences, University of Nottingham, University Park, Nottingham, United Kingdom, 4 Centre for Human Genetics, Katholieke Universiteit Leuven, Leuven, Belgium, 5 Department of Health Technology, Technical University of Denmark, Lyngby, Denmark, 6 European Bioinformatics Institute (EMBL-EBI), Wellcome Genome Campus, Cambridge, United Kingdom, 7 Institute for Clinical Genetics, Faculty of Medicine Carl Gustav Carus, TU Dresden, Dresden, Germany, 8 Department of Human Genetics, University of Leuven, KU Leuven, Leuven, Belgium, 9 Sanger Institute-EBI Single-Cell Genomics Centre, Wellcome Trust Sanger Institute, Hinxton, United Kingdom, 10 Montreal Heart Institute, Université de Montréal, Québec, Canada, 11 European Molecular Biology Laboratory, European Bioinformatics Institute (EMBL-EBI), Wellcome Trust Genome Campus, Hinxton, Cambridge, United Kingdom, 12 Clinic for Pediatric Cardiology—University Hospital of Saarland, Homburg (Saar), Germany, 13 Department of Cellular and Molecular Medicine, University of Copenhagen, Copenhagen, Denmark, 14 Department of Clinical Genetics, Rigshospitalet, Copenhagen University Hospital, Copenhagen, Denmark, 15 Toronto Congenital Cardiac Centre for Adults, and Division of Cardiology, Department of Medicine, University Health Network, Toronto, Canada, 16 Department of Psychiatry, University of Toronto, Toronto, Canada, 17 Department of Pediatrics, Medical College of Wisconsin, Milwaukee, Wisconsin, United States of America, 18 Department of Congenital Heart Disease—Pediatric Cardiology, German Heart Center Berlin, Berlin, Germany, 19 Department of Pediatric Cardiology and Congenital Heart Disease, Heart Center, University of Leipzig, Leipzig, Germany, 20 Department of Pediatric Cardiology, University Hospital Erlangen, Friedrich-Alexander-University Erlangen-Nürnberg (FAU), Erlangen, Germany, 21 Division of Paediatric Cardiology, Royal Brompton Hospital, London, United Kingdom, 22 The Heart Center, Nationwide Children's Hospital, Columbus, Ohio, United States of America, 23 Department of Molecular Genetics, The Ohio State University, Columbus, Ohio, United States of America, 24 Center for Cardiovascular Research, Nationwide Children's Hospital, Columbus, Ohio, United States of America, 25 Department of Pediatrics, The Ohio State University, Columbus, Ohio, United States of America, 26 Competence Network for Congenital Heart Defects, Berlin, Germany, 27 Division of Cardiovascular Sciences, School of Medical Sciences, Faculty of Biology, Medicine and Health, The University of Manchester, Manchester, United Kingdom, 28 Division of Evolution & Genomic Sciences, School of Biological Sciences, Faculty of Biology, Medicine and Health, The University of Manchester, Manchester, United Kingdom, 29 Experimental and Clinical Research Center (ECRC), a joint cooperation between the Charité Medical Faculty and the Max-Delbrück-Center for Molecular Medicine (MDC), Berlin, Germany, 30 Charité—Universitätsmedizin Berlin, corporate member of Freie Universität Berlin, Humboldt-Universität zu Berlin, and Berlin Institute of Health, Department of Pediatric Cardiology, Berlin, Germany, 31 DZHK (German Centre for Cardiovascular Research), partner site Berlin, Berlin, Germany, 32 The Centre for Applied Genomics, The Hospital for Sick Children, Toronto, Canada,

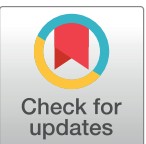

**Data Availability Statement:** All relevant data are within the manuscript and its Supporting Information files. The data used in this study have

been already published. The reference for each individual study is shown in S2 and S3 Tables. The assembled DNV dataset used in this study is provided in the S14 Table. In addition, we have provided a BED file with the CNV dataset (S15 Table).

**Funding:** This study was supported by the German Center for Cardiovascular Research (DZHK) partner sites Berlin, Kiel; the Competence Network for Congenital Heart Defects, National Register for Congenital Heart Defects and KinderHerz (E.A. and M.P.H). BK is supported by a British Heart Foundation Personal Chair (CH/13/2/30154). The funders had no role in study design, data collection and analysis, decision to publish, or preparation of the manuscript.

**Competing interests:** I have read the journal's policy and the authors of this manuscript have the following competing interests: The Department of Molecular and Human Genetics at Baylor College of Medicine receives revenue from clinical genetic testing conducted at Baylor Genetics Laboratories. M.E.H. is a co-founder of, consultant to and holds shares in Congenica, a genetics diagnostic company.

**33** Genome Diagnostics, Department of Paediatric Laboratory Medicine, The Hospital for Sick Children, Toronto, Canada, **34** Department of Internal Medicine, McGovern Medical School, University of Texas Health Science Center at Houston, Houston, Texas, United States of America, **35** Michael E. DeBakey Department of Surgery, Baylor College of Medicine, Houston, Texas, United States of America, **36** Department of Surgery, Division of Cardiothoracic Surgery, Medical College of Wisconsin, Milwaukee, Wisconsin, United States of America, **37** Ruddy Canadian Cardiovascular Genetics Centre, University of Ottawa Heart Institute, Ottawa, Canada, **38** Institute of Human Genetics, University Hospital Ulm, Ulm, Germany, **39** Department of Human Genetics, University Medical Center Schleswig-Holstein (UKSH), Kiel, Germany, **40** Department of Congenital Heart Disease and Pediatric Cardiology, University Heart Center Freiburg—Bad Krozingen, Freiburg, Germany, **41** Department of Medical Biology, Amsterdam UMC, University of Amsterdam, Amsterdam, The Netherlands, **42** Department of Clinical Genetics, Amsterdam UMC, University of Amsterdam, Amsterdam, The Netherlands, **43** Cardiovascular Genetics, Department of Pediatrics, Centre Hospitalier Universitaire Saint-Justine Research Centre, Université de Montréal, Montreal, Canada, **44** Wellcome Sanger Institute, Wellcome Genome Campus, Hinxton, Cambridge, United Kingdom, **45** Laboratory of Translational Genetics, Department of Human Genetics, KU Leuven, Leuven, Belgium

☯ These authors contributed equally to this work.
* larsal@sund.ku.dk (LAL); Marc-Phillip.Hitz@uksh.de (MPH)

## Abstract

Numerous genetic studies have established a role for rare genomic variants in Congenital Heart Disease (CHD) at the copy number variation (CNV) and *de novo* variant (DNV) level. To identify novel haploinsufficient CHD disease genes, we performed an integrative analysis of CNVs and DNVs identified in probands with CHD including cases with sporadic thoracic aortic aneurysm. We assembled CNV data from 7,958 cases and 14,082 controls and performed a gene-wise analysis of the burden of rare genomic deletions in cases versus controls. In addition, we performed variation rate testing for DNVs identified in 2,489 parent-offspring trios. Our analysis revealed 21 genes which were significantly affected by rare CNVs and/or DNVs in probands. Fourteen of these genes have previously been associated with CHD while the remaining genes (*FEZ1*, *MYO16*, *ARID1B*, *NALCN*, *WAC*, *KDM5B* and *WHSC1*) have only been associated in small cases series or show new associations with CHD. In addition, a systems level analysis revealed affected protein-protein interaction networks involved in Notch signaling pathway, heart morphogenesis, DNA repair and cilia/centrosome function. Taken together, this approach highlights the importance of re-analyzing existing datasets to strengthen disease association and identify novel disease genes and pathways.

## Author summary

Congenital heart disease (CHD) is the most common congenital anomaly and represents a major global health burden. Multiple studies have identified a key genetic component contributing to the aetiology of CHD. However, despite the advances in the field of CHD within the last three decades, the genetic causes underlying CHD are still not fully understood. Herein we have assembled a large patient CHD cohort and performed a data-driven meta-analysis of genomic variants in CHD. This analysis has allowed us to strengthen the disease association of known CHD genes, as well as identifying novel haploinsufficient CHD candidate genes.

## Introduction

Congenital Heart Disease (CHD) accounts for a large fraction of foetal and infant deaths, with incidence rates ranging from 7–9 per 1000 live births [1]. Within the last 30 years, survival rates have substantially increased due to improvements in surgical, interventional and clinical intensive care resulting in a rapidly growing number of CHD survivors reaching adulthood [2]. Nevertheless, there is still increased morbidity and mortality in individuals with CHD, resource utilization is high especially among severely affected patients, and importantly, the underlying etiology remains unclear for the majority of cases.

CHD is multifactorial, with both environmental and genetic risk factors [3,4]. Familial aggregation of CHD including Thoracic aortic aneurysm (TAA), as well as a large proportion of genomic copy number variants (CNVs) and *de novo* intragenic variations (DNVs) in probands with CHD suggest a strong genetic component. An estimated 4–20% of CHD cases are due to rare CNVs, suggesting that a significant part of CHD is caused by gene-dosage defects [5]. Recently, exome sequencing in large cohorts has been used to identify novel disease genes and strengthen known disease associations through the demonstration of an excess of *de novo* protein truncating variants (PTV) and rare inherited loss-of-function (LOF) variants in probands with CHD [6,7].

Overlaying both CNVs and PTVs has been used to define novel CHD relevant disease genes in contiguous gene disorders [8,9]. Following this principle, we have performed a genome-wide integrative meta-analysis of published and publicly available datasets of CNVs and DNVs identified in probands with CHD. This analysis, which is one of the larger meta-analyses of genomic variants in CHD so far, strengthens the disease association of known CHD genes and identifies novel haploinsufficient CHD candidate genes.

## Results

### Cohort description and workflow

We assembled a cohort with 7,958 cases (comprising both non-syndromic CHD, syndromic CHD and TAA cases) and 14,082 controls (**S1 Table**). Of the total of cases, 777 (~10%) were diagnosed with Thoracic Aortic Aneurysm (TAA). An overview of the sources used to assemble the present cohort is listed in **S2 Table** (for CHD cases) and **S3 Table** (for controls). We applied a set of quality control filters to our assembled CNV data before performing case-control association tests (Materials and Methods). In addition, common CNVs (minor allele frequency (MAF) in controls > 0.01) were excluded from the analysis. After filtering, 6,746 cases and 14,024 controls remained for further downstream analysis. Furthermore, we built a dataset of *de novo* variations (DNVs) identified in 2,489 probands with CHD from parent-offspring trios [6,7].

### CNV burden test of known CHD genes

Haploinsufficiency has been shown to cause a reasonable proportion of CHD [5]. Thus, genes known to be associated with CHD and genes which are intolerant for LOF variations should be deleted more often in probands with CHD than in controls. To test this hypothesis, we performed a CNV burden test using sets of genes known to be involved in CHD. In addition, we included genes known to be associated with developmental disorders, a curated list of known haploinsufficient disease genes, autosomal recessive disease genes and genes predicted to be intolerant to LOF variations (based on the observed/expected LOF ratio from gnomAD [10]). The burden test was performed using a logistic regression framework [11] (implemented in

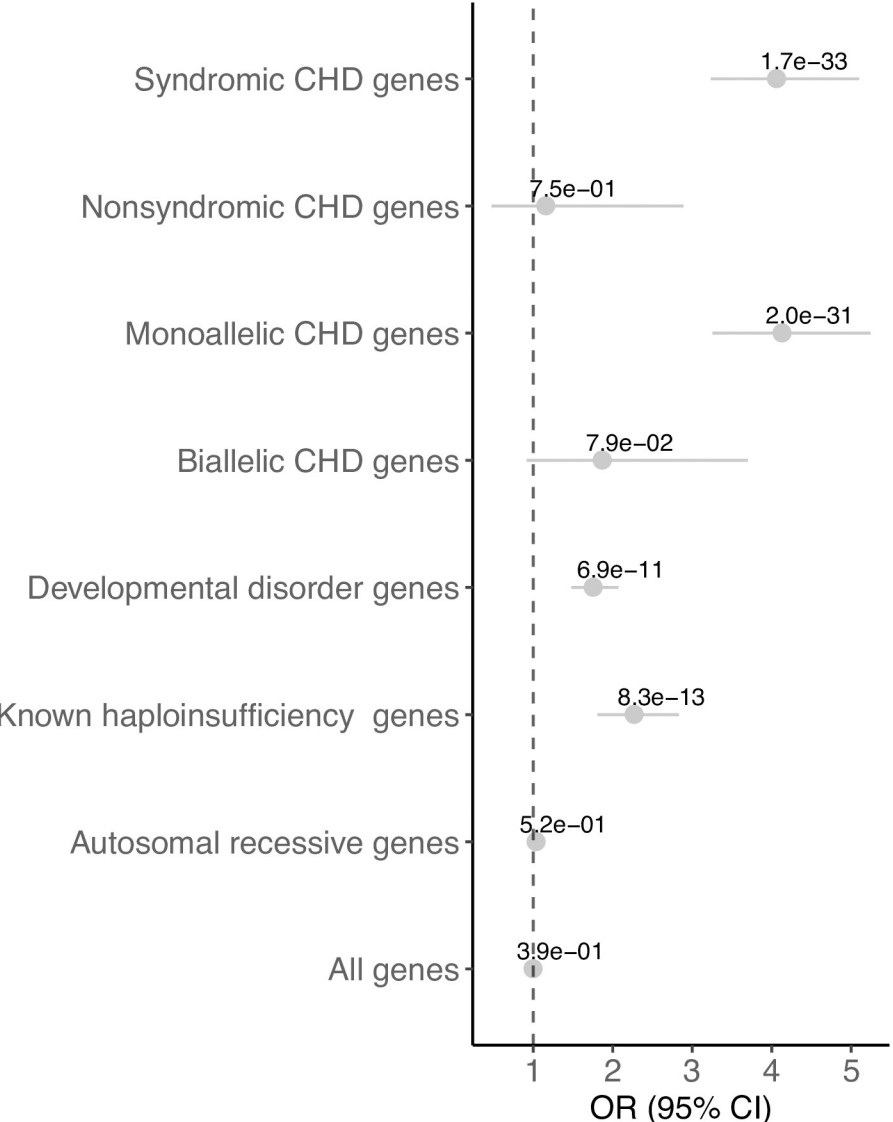

**Fig 1. CNV burden test on known gene sets.** The forest plot shows the odds ratio (dots), the 95% confidence intervals indicating the certainty about the OR (interrupted line) and the *P-value* in the indicated gene sets.

PLINK v1.7). **Fig 1** and **S4 Table** summarize the results from the burden test on the different gene sets: known CHD genes (grouped in syndromic, non-syndromic, monoallelic and biallelic), developmental disorder genes, haploinsufficiency disease genes, autosomal recessive genes and all protein-coding genes. We tested all protein-coding genes to address the possibility that the analyses could be biased by differences in the CNV rate within the case and control groups, since we have assembled our cohort from different datasets. We did not observe genome-wide (all tested protein-coding genes) enrichment ($P = 0.39$, $OR = 0.99$) nor enrichment in the autosomal recessive gene set ($P = 0.52$, $OR = 1.03$) when comparing rare CNV deletions in cases vs controls. In contrast, the analysis revealed significant differences in the burden of CNV deletions between cases and controls for the set of haploinsufficiency genes ($P = 8.29 \times 10^{-13}$, $OR = 2.27$). As expected, our analysis revealed significant enrichment for the set of known

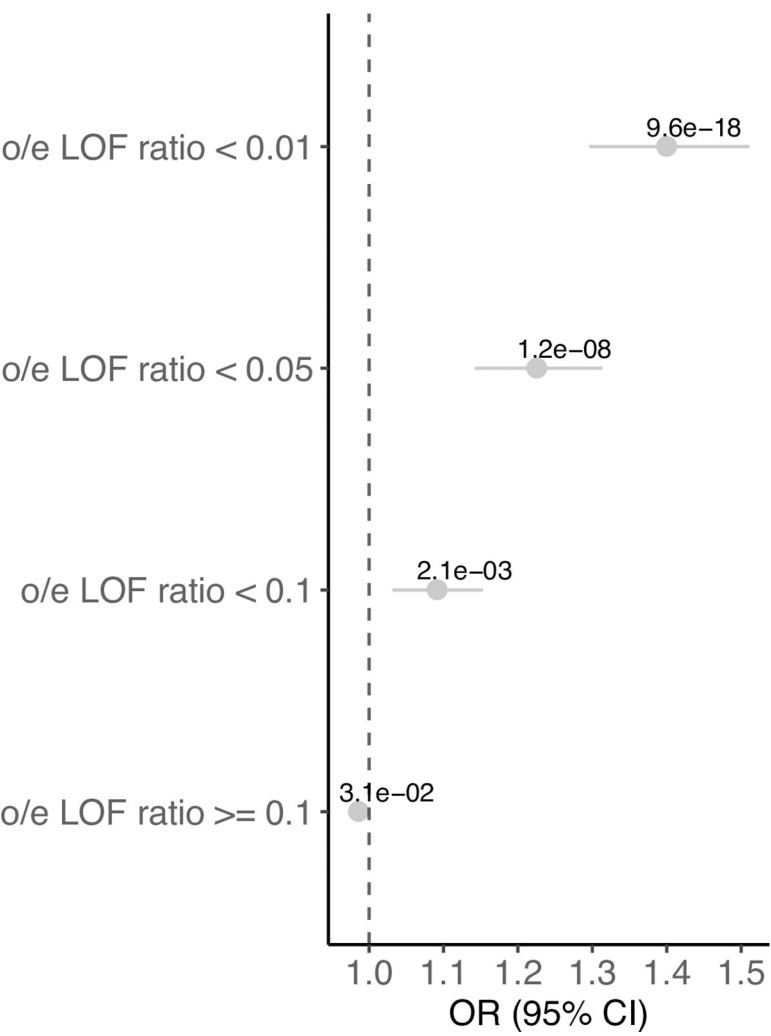

**Fig 2. CNV burden test on constraint LOF genes at different observed/expected LOF ratio thresholds.** The forest plot shows the odds ratio (dots), the 95% confidence intervals indicating the certainty about the OR (interrupted line) and the *P-value* in the indicated gene sets.

CHD genes, which is mainly explained by the contribution of monoallelic CHD genes ($P = 2.04 \times 10^{-31}$, $OR = 4.13$) and syndromic CHD gene set ($P = 1.66 \times 10^{-33}$, $OR = 4.06$). Unlike the monoallelic and syndromic CHD gene sets, no significant enrichment was found for the nonsyndromic ($P = 0.75$, $OR = 1.16$) and biallelic ($P = 0.08$, $OR = 1.87$) CHD gene sets. Our analysis revealed a moderate enrichment of rare CNVs in the developmental disorder gene set ($P = 6.90 \times 10^{-11}$, $OR = 1.75$).

When the regression-based analysis was performed at different levels of the observed/expected LOF ratio (*oeLOF*) constraint metric (**Fig 2** and **S5 Table**), we observed the higher enrichment toward the most LOF constrained genes (*oeLOF* $< 0.01$, $P = 9.55 \times 10^{-18}$, $OR = 1.40$) and still a moderate enrichment for genes with *oeLOF* $< 0.1$ ($P = 0.002$, $OR = 1.09$). No enrichment was observed in the set with *oeLOF* ratio $> = 0.1$ ($P = 0.03$, $OR = 0.99$). Based on these results we conclude that haploinsufficiency causes a significant component of CHD.

## Genome-wide identification of haploinsufficiency candidate disease genes for CHD

To perform a systematic, genome-wide identification of potential haploinsufficient CHD disease genes and loci, we analysed the CNV burden of 19,969 protein-coding genes (GENCODE v19). To this end, we compared the number of rare CNV deletions (MAF < 0.01) among cases and controls for each gene, and identified genes with significant CNV burden using a permutation test (significance level of adjusted $P < 0.05$, see Materials and Methods). If a CNV spanned two or more genes, all affected protein-coding genes were considered in the analysis. The distributions of rare CNV deletions in CHD cases across all 22 human autosomes is shown in **Fig 3**. Significant candidate genes had a median number of 12 overlapping CNVs in cases, compared to a median of 0 overlapping CNVs in controls (**S1A Fig**). Because CNVs can be large chromosomal aberrations, multiple genes were affected by some of the CNVs. In total, 528 genes (Sheet A in **S6 Table**) reached significance (Permutation test, $P$ adjusted < 0.05). These 528 genes encompass a total of 63 loci (Sheet B in **S6 Table**, highlighted in magenta in

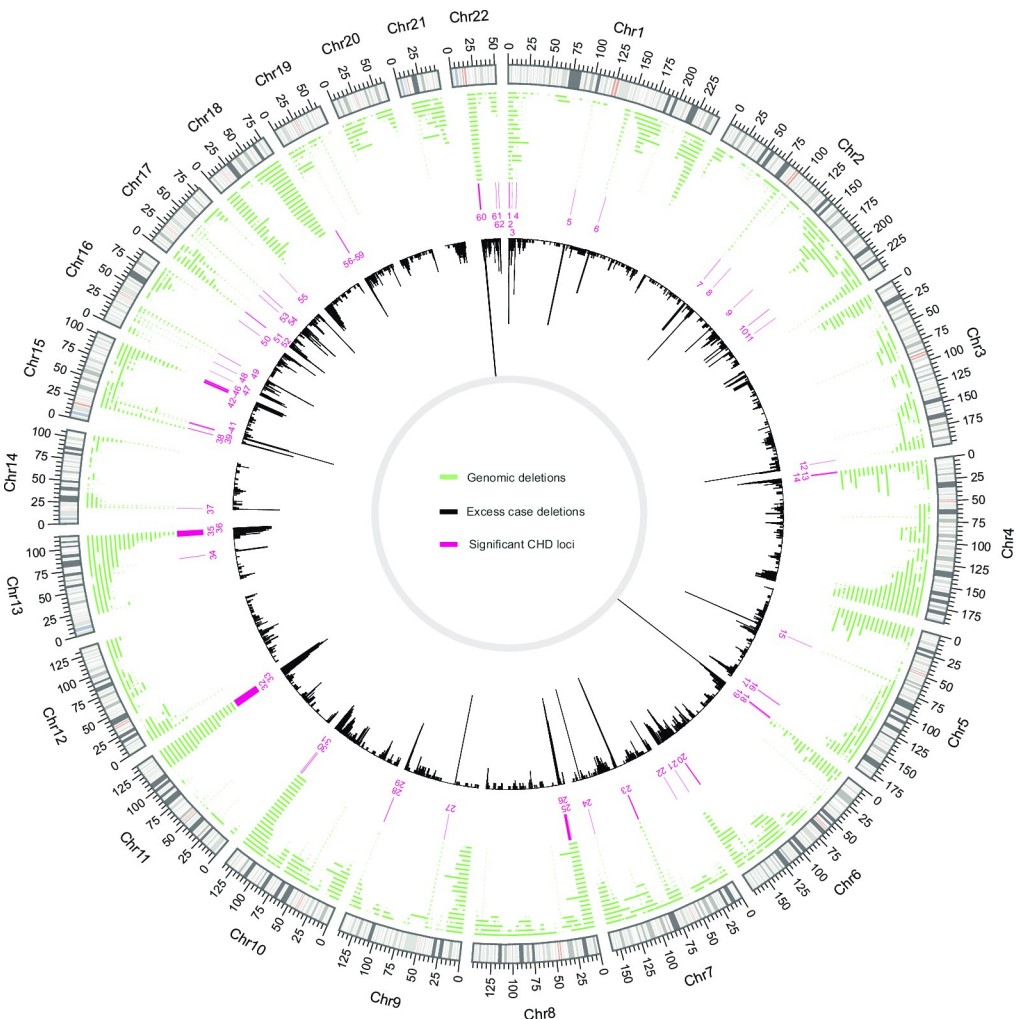

**Fig 3. CNV deletion distribution across the 22 autosomes.** The plot shows the distribution of rare CNV deletions (green track) in CHD cases, the differences between the overlapping CNV deletions in cases and controls (black track) and highlight the location of the 63 significant loci discovered (in magenta).

Fig 3). The sizes of these loci range from 558 bp to 10.5 Mbp, with a median value of 243 Kbp (S1B Fig). The number of genes per locus ranged from 1 to 48, with a median value of 3 (S1C Fig). Only 16 loci contained a single gene (Sheet B in S6 Table).

In addition, we tested previously described CNV deletion syndrome regions ([https://decipher.sanger.ac.uk/disorders#syndromes/overview](https://decipher.sanger.ac.uk/disorders#syndromes/overview)) associated with developmental disorders and/or CHD for enrichment in our analysis (Materials and Methods). We found eight of these regions enriched in the dataset (S7 Table), with the 16p11.2-p12.2 locus being the region with the largest number of deletions in cases (n = 230).

## Shared genetic architecture of CHD and TAA

We independently performed a genome-wide test without the TAA cases to evaluate its impact on CHD. As expected, most of the genes (447 out of 528) remained significant after removing the contribution of TAA cases, since ~90% of the cases in the analyzed CNV cohort were CHD. Ten genes were significantly enriched independently when analyzing CHD and TAA cases, while 61 were significantly enriched only in TAA cases (S8 Table).

## De novo variation analysis

To identify an independent set of haploinsufficient CHD candidate genes, we combined *de novo* variations identified in two large-scale CHD case-control studies [6,7] and performed a gene-based *de novo* variation (DNV) burden test [12]. We analysed a total of 4,195 rare DNVs within 2,534 genes in the patient cohort. After classifying every variant into functional groups (Materials and Methods) 526 of these variants were predicted to be protein-truncating and 2,647 were missense. We evaluated for potential differences of the DNV rates between cohorts (see Materials and Methods). Comparison of the rate of each variant type across the groups was non-significant ($P > 0.05$, Poisson test, S9 Table).

We used two available statistical methods, Mupit [12] and DeNovoWEST [13], which test the significance of observed DNV at gene level, by comparing the number of observed variations with the number of expected variations (based on a sequence-dependent variation recurrence rate, see Materials and Methods). While Mupit focuses on enrichment of protein-truncating DNVs specifically, the DeNovoWEST test incorporates missense constraint information at variant level and applies a unified severity scale at variant level based on the empirically-estimated positive predictive value of being pathogenic. Based on the complementary results of both tests [13], we reported the minimal observed DNV *p-value* ($P_{dnv}$) per gene.

We identified 14 genes significantly enriched in the DNV analysis ($P < 0.05$ after Bonferroni correction for multiple testing, S10 Table). All of these genes were affected by at least two constrained non-synonymous DNV (nsDNV) and show significant overlap with 11/14 (78.6%) of the genes being known CHD disease genes. *CHD7* (OMIM 214800) was the most significant haploinsufficient gene ($P = 2.84 \times 10^{-26}$) with 18 nsDNVs identified in the patient cohort. Other highly enriched genes for nsDNV—*KMT2D* (OMIM 147920), *KMT2A* (OMIM 605130), *NSD1* (OMIM 117550), *TAB2* (OMIM 614980), and *ADNP* (OMIM 615873)—have been previously associated with different types of neurodevelopmental disorders with co-occurrence of CHD. In the case of *KDM5B* (OMIM 618109), it has only been described in the context of a recessive neurodevelopmental phenotype with cases presenting ASD (Atrial septal defects) [14,15].

We next evaluated the distribution of o/e LOF ratio at different levels of DNV enrichment (genes were split based on $P_{dnv}$). Since the o/e ratio of LOF variation in each gene is strongly affected by its length, we instead used the 90% upper bound of its confidence interval (termed LOEUF), which keeps the direct estimate of the o/e ratio and allows to distinguish small genes

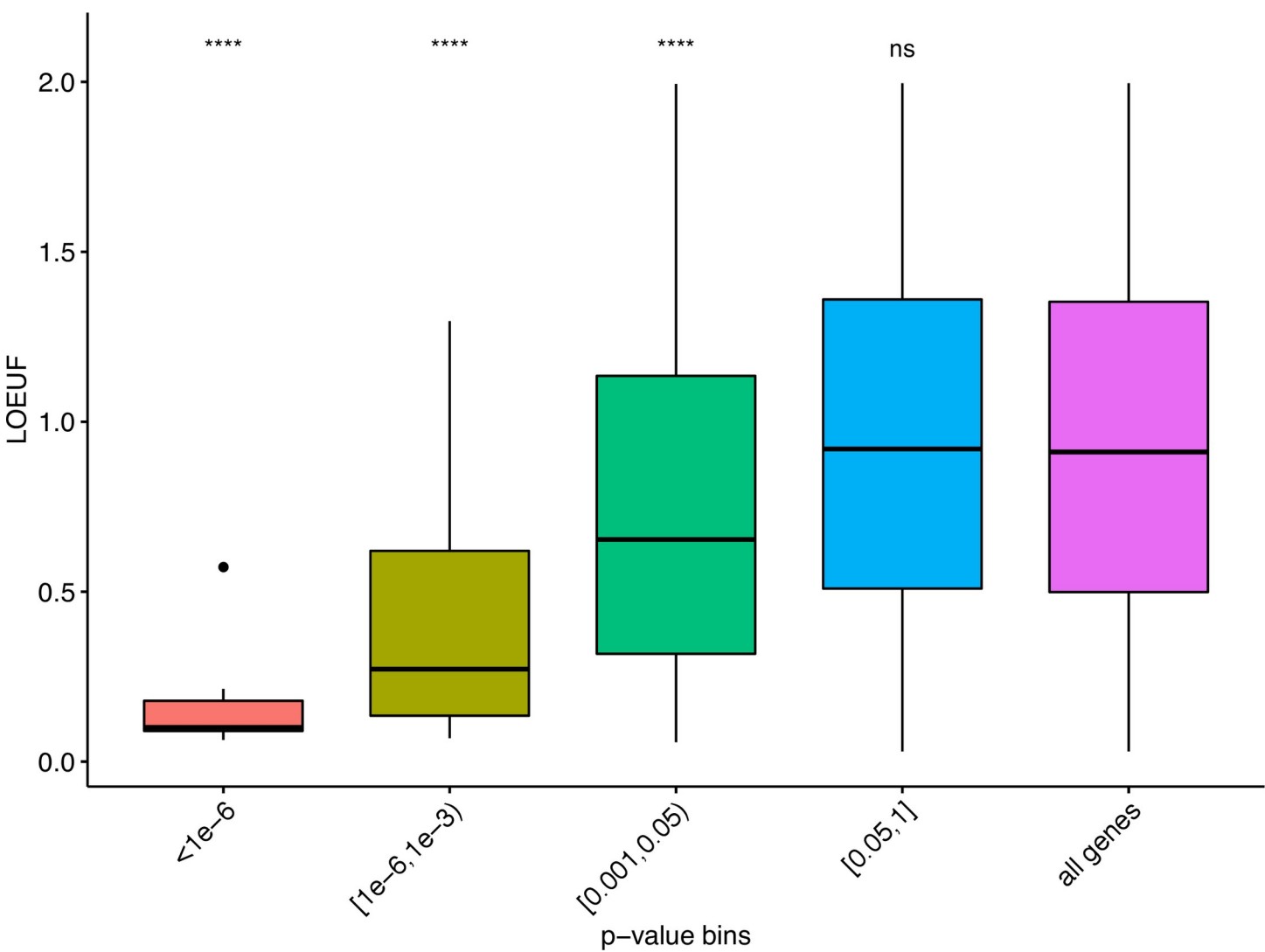

**Fig 4. Comparison of the distribution of LOEUF metric at different level of significance of nsDNV-enriched genes.** X-axis denotes the P-values from the DNV analysis (binned). Y-axis denotes the o/e LOF ratio upper bound fraction (LOEUF). All groups were compared against the LOEUF distribution of all protein-coding genes (purple). Differences between the distributions were tested using a two-sided Wilcoxon rank sum test. ****: P<0.0001, ns: non-significant.

from large genes, as suggested by Karczewski *et al* [10]. We observed that the genes with higher enrichment for nsDNV (lower $P_{dnv}$) show a significant decreased LOEUF compared to the mean of all protein-coding genes (**Fig 4**).

## Integration of DNV and CNV results

To identify high confidence haploinsufficient CHD disease genes, we performed a joint analysis integrating the results from the CNV and the DNV analysis. We combined the results from both analyses ($P_{dnv}$ and $P_{cnv}$) using the Fisher combine method. We demonstrated that both enriched genes for DNV and CNV deletions are significantly represented among LOF constraint genes (measured by the o/e LOF ratio). Therefore, we applied a Bonferroni multiple testing correction using independent hypothesis weighting [16] (IHW) by incorporating the gene o/e LOF ratio, as a measure of haploinsufficiency (**S2 Fig**). Our analysis revealed 21 genes that were significantly enriched for CNV deletions and/or non-synonymous DNV (**Table 1**).

**Table 1. Top 21 significant genes arising from both the permutation-based test and the DNV rate-based test.** Cases/Controls: Number of cases and controls carrying CNV deletions overlapping the gene in the CNV analysis. $P_{cnv}$: p-value from the CNV permutation test. nsDNV: Number of constrained non-synonymous variations in the *de novo* analysis. $P_{dnv}$: p-value from the DNV analysis. Significant: The analysis where the gene was significant (dnv: DNV analysis, cnv: CNV analysis, both: Both analysis, none: Non-significant neither DNV nor CNV analysis). *metaP*: combined p-value ($P_{dnv}$ and $P_{cnv}$) using the Fisher method. $P_{ihw}$: Bonferroni corrected p-value using independent hypothesis weighting (IHW) and LOEUF metric as covariate. LOEUF: o/e LOF ratio upper bound fraction from gnomAD. *All the 21 genes were significant after combining their p-values and applying Bonferroni correction. [1]Evidence is from mouse models [24,62].

| | CNV | | | DNV | | | Combined | | | |
|---|---|---|---|---|---|---|---|---|---|---|
| Gene | case | control | $P_{cnv}$ | nsDNV | $P_{dnv}$ | *Significant | *metaP* | $P_{ihw}$ | LOEUF | Known CHD |
| CHD7 | 6 | 1 | 6.80E-03 | 18 | 2.84E-26 | dnv | 1.25E-26 | 8.05E-23 | 0.076 | Yes |
| KMT2D | 0 | 0 | 1.00E+00 | 18 | 1.32E-25 | dnv | 7.67E-24 | 4.93E-20 | 0.103 | Yes |
| NSD1 | 1 | 1 | 5.63E-01 | 12 | 1.00E-14 | dnv | 1.90E-13 | 2.14E-09 | 0.095 | Yes |
| KMT2A | 0 | 0 | 1.00E+00 | 7 | 1.00E-14 | dnv | 3.32E-13 | 1.86E-09 | 0.065 | Yes |
| NOTCH1 | 10 | 24 | 1.00E+00 | 7 | 1.00E-14 | dnv | 3.32E-13 | 2.14E-09 | 0.097 | Yes |
| TAB2 | 12 | 0 | 1.00E-04 | 5 | 3.46E-09 | both | 1.03E-11 | 5.75E-08 | 0.098 | Yes |
| ANKRD11 | 13 | 0 | 1.00E-04 | 3 | 2.32E-05 | cnv | 4.85E-08 | 2.72E-04 | 0.107 | Yes |
| WHSC1 | 11 | 0 | 1.00E-04 | 3 | 8.73E-05 | cnv | 1.71E-07 | 9.96E-04 | 0.119 | No[1] |
| ADNP | 0 | 0 | 1.00E+00 | 4 | 9.94E-09 | dnv | 1.93E-07 | 1.13E-03 | 0.123 | Yes |
| DYRK1A | 4 | 0 | 1.43E-02 | 4 | 9.46E-07 | dnv | 2.59E-07 | 1.64E-03 | 0.214 | Yes |
| NALCN | 10 | 1 | 1.00E-04 | 3 | 1.76E-04 | cnv | 3.32E-07 | 6.83E-03 | 0.522 | No |
| ELN | 30 | 0 | 1.00E-04 | 2 | 1.77E-04 | cnv | 3.34E-07 | 7.50E-03 | 0.871 | Yes |
| WAC | 7 | 0 | 4.00E-04 | 3 | 1.31E-04 | none | 9.33E-07 | 5.44E-03 | 0.084 | No |
| RBFOX2 | 1 | 0 | 3.45E-01 | 4 | 1.59E-07 | dnv | 9.72E-07 | 6.25E-03 | 0.194 | Yes |
| KANSL1 | 94 | 110 | 2.00E-04 | 2 | 3.38E-04 | none | 1.19E-06 | 6.92E-03 | 0.238 | Yes |
| MYO16 | 13 | 2 | 1.00E-04 | 2 | 8.38E-04 | cnv | 1.45E-06 | 1.74E-02 | 0.272 | No |
| MED13L | 2 | 1 | 2.70E-01 | 4 | 4.58E-07 | dnv | 2.09E-06 | 1.22E-02 | 0.064 | Yes |
| KDM5B | 0 | 0 | 1.00E+00 | 4 | 1.45E-07 | dnv | 2.43E-06 | 4.97E-02 | 0.572 | No |
| GATA6 | 0 | 0 | 1.00E+00 | 5 | 1.80E-07 | dnv | 2.98E-06 | 1.92E-02 | 0.174 | Yes |
| ARID1B | 4 | 0 | 1.31E-02 | 4 | 1.48E-05 | none | 3.20E-06 | 1.87E-02 | 0.102 | No |
| FEZ1 | 10 | 0 | 1.00E-04 | 2 | 2.22E-03 | cnv | 3.62E-06 | 4.30E-02 | 0.414 | No |

A gene was included in the final set of haploinsufficient CHD disease genes if it reached a significant corrected *metaP* < 0.05 (after Bonferroni adjustment with IHW).

## Subclassification of CHD phenotypes

We performed a further analysis based on specific CHD subtypes in addition to the collective analysis of all CHD phenotypes. The analysis focused on simplex CHD cases, within two main categories: Conotruncal ($N_{CNV}$ = 873, $N_{DNV}$ = 234) and LVOTO ($N_{CNV}$ = 594, $N_{DNV}$ = 351). We only included cases with a clear phenotypic description and without any overlapping phenotypic features between the two categories (LVOTO/Conotruncal). The conotruncal group consisted mostly of Tetralogy of Fallot (TOF), Truncus arteriosus and Transposition of great arteries (TGA), whereas LVOTO mainly constituted Aortic stenosis (AS) including Bicuspid aortic valve disease (BAV), Coarctation and Hypoplastic left heart syndrome (HLHS).

As described above, we performed the same integrative approach for the LVOTO and conotruncal groups to identify CHD subtype-specific genes. Our analysis showed four significant genes (**S11 Table**). Three are observed in LVOTO (*KMT2D*, *KMT2A* and *TAB2*) and a single gene showed significant enrichment in the conotruncal subtype (*NSD1*).

## Significant CHD genes are highly and/or differentially expressed in the heart

We next evaluated the expression pattern of the 21 significant genes (Bonferroni corrected *metaP* < 0.05) in the heart using RNA-Seq data from human tissues at different developmental

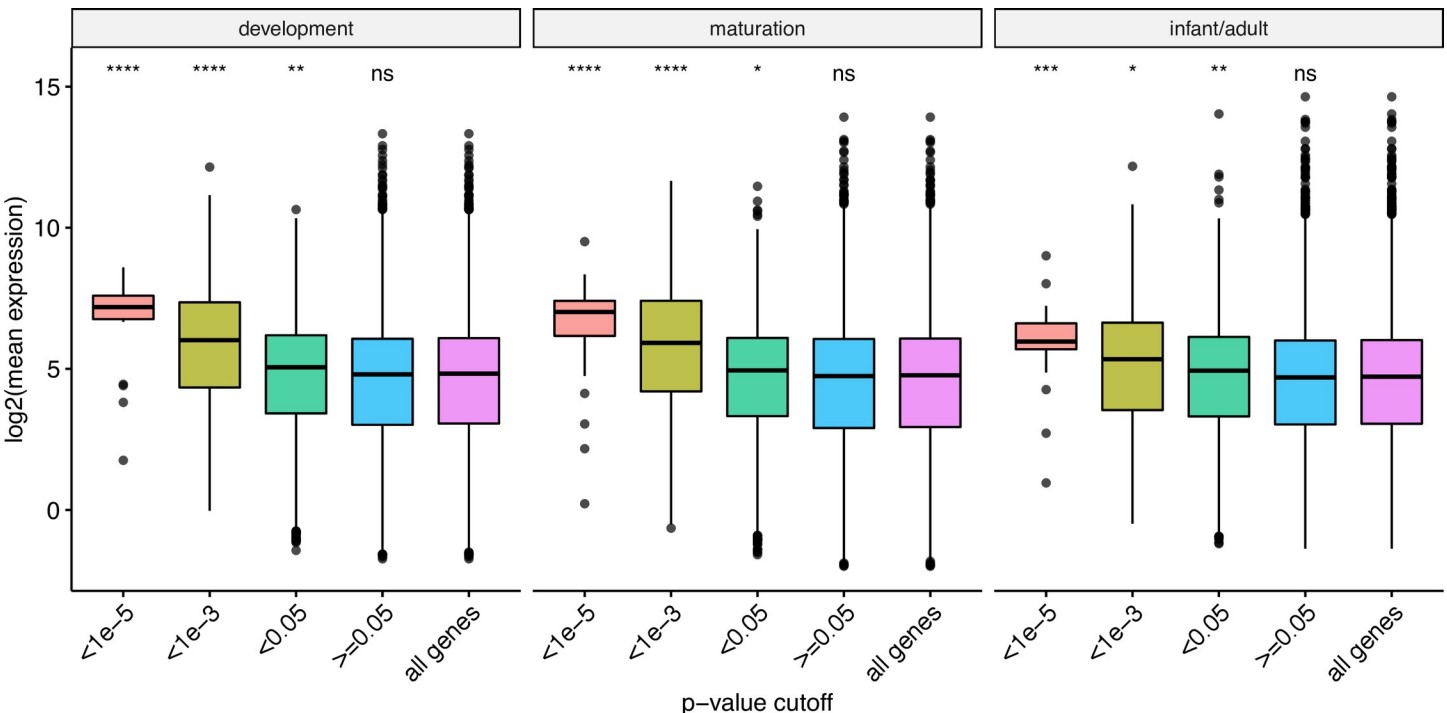

**Fig 5. Comparison of the mean expression (heart) distribution at different *metaP* cut-offs.** Panels show three different heart development stages: early development, maturation and infant/adult. X-axis denotes the combined p-value from DNV and CNV analysis (*metaP*, at different cut-offs). Y-axis denotes the genes' mean expression in the heart (log scale). The 21 significant candidate CHD genes (**Table 1**) are contained in the fraction with the higher expression (red box). Differences between the distributions were tested using a two-sided Wilcoxon rank sum test (reference group: all genes). ****: P<0.0001, ***: P<0.001, **: P<0.01, *: P<0.05, ns: non-significant.

time points [17]. We stratified the analysis based on stages of heart development (see Materials and Methods). Our analysis revealed that the most significant genes (*metaP* < 1 x 10e⁻⁵) show significantly increased mean expression in the heart ($P < 0.0001$, Wilcox test) at different developmental stages (development, maturation and infant/adult), compared to all protein-coding genes (**Fig 5**). Moreover, 18 out of 21 genes fall in the in the top quartile of heart expression in both developmental and maturation stages (**S3 Fig**). To complement our expression analysis, we compared gene expression during human heart development with expression in two other mesodermal organs: kidney and liver. This allowed identification of genes with significant changes in its expression levels during crucial heart developmental stages, which would have not been possible when focusing on expression levels alone (Materials and Methods). We found that 17 out of 21 CHD candidate genes are differentially expressed in the heart ($R^2 > 0.50$, Bonferroni corrected $P < 0.01$) when compared to its expression levels in kidney and/or liver. Interestingly, the three genes (*FEZ1*, *NALCN* and *MYO16*) which are not among the highly expressed genes, were found to be significantly differentially expressed during heart development compared to kidney and/or liver (**S12 Table**).

## CNV/DNVs burden of specific protein complexes

CNVs and DNVs can affect heart development either through haploinsufficiency of a single gene, or through its combined impact on the function of several genes. Indeed, oligogenic models have been implicated in CHD, and proteins acting in the same complex or pathway are known to be encoded in genomic clusters [18,19]. We therefore conducted a systems-level analysis to identify global mechanisms by which haploinsufficiency might promote CHD. In

particular, we assessed the combined effect of CNVs and DNVs with respect to human protein-protein interactions (PPIs). The InWeb and ConsesusPathDB databases provides ranked information about experimentally determined physical interactions and, therefore, serves as a proxy to understand the functional effects of CNV/DNVs on human protein complexes (Materials and Methods). The genes with Benjamini–Hochberg adjusted *metaP* < 0.05 (n = 492 genes) were used as seeds to build a PPI network from the data available in InWEb and ConsessusPathDB. No additional interections were considered. The final network consisted of 164 proteins and 290 interactions (**S4 Fig**). A total of 10 overlapping sub-clusters within this network were identified using the in-built clustering algorithm implemented in GeNets [20] (Materials and Methods). Gene-ontology (GO) enrichment analysis suggested that four out of these ten sub-clusters are enriched for genes involved in Notch signaling pathway, cardiocyte differentiation, DNA repair and centrosome function (**Fig 6**). All the four clusters accommodate more CNV deletions in CHD cases compared to controls. Six out of the ten sub-clusters did not show significant enrichment for any particular biological process.

## Discussion

We performed a meta-analysis of rare genomic variants in a cohort of 10,447 CHD probands, which provides a useful resource for interpreting CNVs and DNVs identified in patients with CHD. We implemented a statistical approach which allows the integration of different types of genomic variants to discover novel genes associated with CHD. Our data-driven integrative analysis took into account three major criteria at the genomic level: a) gene enrichment for DNVs, b) gene enrichment for CNV deletions and c) gene intolerance for LOF variations. Our analysis identified 21 significant haploinsufficient CHD genes. Fourteen of these are known CHD genes, and the remaining seven genes have not previously been associated with CHD (**Table 1**).

To further strengthen associations, we made use of a newly published human transcriptome atlas covering different developmental, maturation and adult stages in numerous organs [17]. Similar to previous results [7], our analysis highlights that the majority of the 21 significant genes are highly expressed during critical stages of heart development. Unlike earlier studies [7,21] which did not address the importance of expression changes over time, we evaluated the differential expression patterns of genes by comparing levels of expression in the heart, kidney and liver at different time points in development. This analysis allowed us to strengthen disease association for genes not falling under the high expression group and highlight the critical importance of all 21 genes independently of the genomic approach. This aspect is complemented by the fact that the majority of genes (14/21) were already known to cause CHD. To further strengthen disease associations, spatiotemporal expression at single-cell resolution during critical cardiac developmental timepoints and analyses of animal models with targeted mutation in the candidate disease genes is warranted. This could strengthen disease association further and provide pathophysiological information.

Among the 21 likely haploinsufficient disease genes for which the combined analyses showed enrichment (Bonferroni corrected *metaP* < 0.05), 14 genes (*CHD7, KMT2D, KMT2A, NOTCH1, NSD1, TAB2, ANKRD11, ADNP, DYRK1A, RBFOX2, KANSL1, ELN, MED13L* and *GATA6*) are well-established CHD genes, and our data confirms this association. To the best of our knowledge, association between CHD and seven genes (*KDM5B, WHSC1, WAC, NALCN, ARID1B, FEZ1* and *MYO16)* had either not been established, or had been reported in small cases studies or a single individual only.

*KDM5B* is not an established CHD gene thus far, although one patient with compound heterozygous frameshift variants had an ASD [14]. While some have argued against the

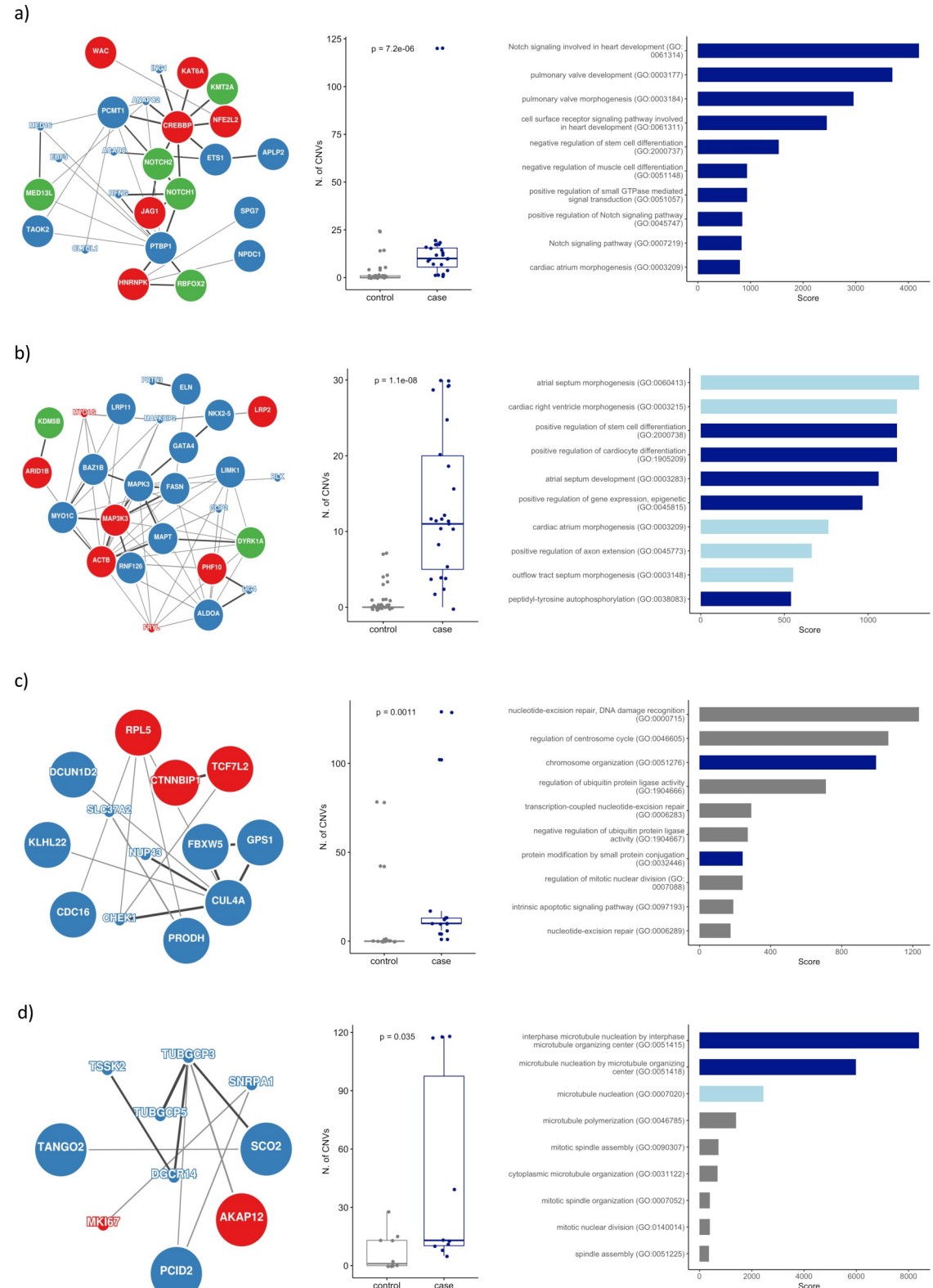

**Fig 6. Identification of functional networks enriched for proteins encoded by genes affected by CNVs and/or DNVs associated with CHD.** The protein-protein interaction networks (a-d, for clusters 1, 3, 8 and 9 respectively) were identified using GeNets (**S4 Fig**). Proteins are shown as nodes, interactions as edges. Enrichment for CNVs (blue) and DNVs (green) are highlighted. Proteins

with no specific enrichment for CNV and/or DNVs but with B-H adjusted *metaP* < 0.05 are highlighted in red. The size of the circles denotes if the genes was found significantly highly and/or differentially expressed in the heart (large circles: significant expression; small circles: non-significant). The distribution of CHD case-CNVs and control-CNVs are shown for each cluster. Significant difference in the CNV distribution was calculated using a Wilcox rank sum test. The horizontal bar plots show the top ten GO enriched terms for each cluster (output from Enrichr tool). X-axis in the horizontal bar plot denotes the combined score from Enrichr, which is computed by multiplying the log-transformed p-value and the z-score. Bar color encoded the GO biological process significant level (dark blue: FDR < 5%, light blue: FDR 5–10%, grey: FDR > 10%).

haploinsufficiency of the gene [22], our analysis suggests *KDM5B* as a plausible haploinsufficient CHD gene. Additional functional studies are warranted to confirm its role in CHD.

A recent CNV meta-analysis [23] based on non-syndromic CHD patients found that duplication of *WHSC1* (also known as *NSD2*) is a possible cause of CHD. However, haploinsufficiency of *WHSC1* has not previously been associated with CHD. In support of its role in CHD, *Whsc1* has been reported to cause heart malformations in mouse models [24]. In addition, *WHSC1* is known to interact with *NKX2.5* [24]. In spite of this, the low incidence of CHD in individuals with Wolf-Hirschhorn syndrome suggests that haploinsufficiency of *WHSC1* alone does not cause CHD.

Heterozygous truncating variations in *WAC*, as well as CNV deletions involving this gene, have been recently associated with the DeSanto-Shinawi syndrome, a rare neurodevelopmental disorder characterized by global developmental delay [25,26]. Furthermore, in two non-consanguineous unrelated individuals with heart malformations, among other disorders [27], microdeletions at 10p11.23-p12.1 (overlapping *ARMC4*, *MPP7*, *BAMBI* and *WAC*) were identified. Despite these isolated reports, no definite association between *WAC* and CHD has been established.

DNVs in *NALCN* have been reported to cause a dominant condition characterized by multiple features including developmental delay, congenital contractures of the limbs and face and hypotonia [28,29]. However, among the phenotypes observed, CHD have been not described thus far.

Heterozygous variation of *ARID1B* is a frequent cause of intellectual disability [30,31]. A recent analysis of 143 patients with *ARID1B* variations showed that individuals display a spectrum of clinical characteristics. Congenital heart defects were observed in 19.5% of the patients [32].

*FEZ1* is a neurodevelopmental gene, which has been associated with schizophrenia [33]. *Fez1* has been reported to be regulated by *Nkx2-5* in heart progenitors in mice, suggesting a possible role in heart development [34].

*MYO16* (*NYAP3*) encodes an unconventional myosin protein, involved in regulation of neuronal morphogenesis [35]. We have not found an association between *MYO16* and heart development in the literature.

Although, several genes have been shown to be altered in syndromic and non-syndromic cases with CHD and TAA (e.g. *HEY2* [36], *MYH11* and *NOTCH1* [37]), among the 10 genes significant in our analysis for TAA, CHD and the combined scenario, none has been reported previously to be associated with either CHD or TAA. Given the limited data size and only accessing CNV calls from TAA cases, future studies looking at CNV and DNV in both phenotypes are required to establish stronger genotype-phenotype correlation to better understand a possibly shared genetic architecture for the two disease entities.

Also, our study did not identify strong signals associated with specific CHD subtypes. We identified four genes with significant enrichment (adjusted *metaP* < 0.05) within the two evaluated CHD subtypes (LVOTO and conotruncal defects). Three were associated with LVOTO, and the contribution was mainly from DNV. *KMT2D* was enriched in LVOTO, which is

consistent with the reported spectrum of CHD in patients with Kabuki syndrome, where a large proportion of individuals have LVOTO type CHD [38,39]. PDA and septal defects predominate in Wiedemann-Steiner Syndrome (*KMT2A*). However, aortic insufficiency and BAV have all been reported [40] suggesting that LVOTO might form part of the phenotypic spectrum. *TAB2* was also enriched in LVOTO. Mutations in *TAB2* are associated with a wide range of cardiac phenotypes [41–43]. Deletions at 6q24 causing haploinsufficiency of *TAB2* have been associated with outflow tract abnormalities, including LVOTO [44,45], which might point to a role of *TAB2* LVOTO pathogenesis. *NSD1* was the only gene identified as significant among conotruncal defects. The significance of this is unclear as septal defects predominate in Sotos Syndrome. However, patients in previous studies were ascertained focusing on Sotos syndrome (OMIM 117550) rather than CHD. Given the current size of each CHD subgroup and the low number of CNV and DNV events in these genes, these results cannot be considered conclusive. More precise phenotypic descriptions for each individual are necessary to increase the genotype-phenotype correlation for individual genes and improve the identification of genetic subnetworks, along with larger sample sizes.

In addition to the gene-centered analysis, we also applied a systems-level analysis in order to identify potential novel pathophysiological mechanisms affected by haploinsufficiency. In this approach, we took advantage of GeNets [20], a computational framework for the analysis of protein-protein interactions, developed for the interpretation of genomic data. Our analysis allowed us to identify PPI clusters enriched for genes affected by CNVs and/or DNV in patients. Furthermore, GO enrichment analysis suggested distinct biological functions for four of these clusters.

Cluster 1 (**Fig 6A**) contains proteins involved in the Notch signaling pathway. Our data corroborate previous studies that confirm the central role of Notch pathway in the pathophysiology of CHD [46] and highlights the shared contribution of CNVs and DNVs within the cluster. Cluster 3 (**Fig 6B**) contains proteins driving essentials processes in the development of the heart such as atrial septum and cardiac right ventricle morphogenesis as well as proteins playing significant role in the positive regulation of gene expression. These mechanisms has been well studied elsewhere [47]. Interestingly, three out of the seven candidate novel CHD genes (*WAC*, *ARID1B* and *KDMB5*) were found to be contributing to these two clusters. Cluster 8 (**Fig 6C**) showed enrichment for processes related with chromosome organization and DNA repair. Association between DNA repair and CHD is not well established thus far. Cluster 9 (**Fig 6D**) was found to be associated with microtubule organizing function. This biological process has been not described in the context of CHD, although an earlier report [48] describes complex CHD among the phenotypes in individuals with 15q11.2 deletion syndrome, which involves the tubulin gamma complex protein 5.

Given the heterogenous data sources and the complex inheritance patterns often observed in patients with CHD, our study has limitations. Firstly, the patient data was collected from almost 200 different published sources, and in many cases it was only possible to obtain data from CNV calls which had already been suggested to be pathogenic. Thus, we are aware that our patient data are incomplete because genome-wide CNV data are missing from a large part of the patient cohort and re-emphasizes already established associations. This is not the case for controls, for which genome-wide data was used. As a direct consequence, even though the difference between the rates of CNV deletions in controls and cases decreased dramatically after applying a quality control filtering step, a slight difference remained between both cohorts. The lack of collected CNV data spanning sex chromosomes limited the analysis only to autosomal chromosomes. In addition, the distribution of CNVs that overlap known microdeletion syndromes such as DiGeorge syndrome and Williams syndrome is overrepresented in the dataset. Similarly, the degree of phenotyping varied across the different studies, and

often only basic phenotypic terms relating to CHD were available. This made it impossible to refine the diagnosis to a precise phenotypic class of CHD in many individuals.

Previous research has shown that the chances of finding a genetic cause of CHD is higher in syndromic, rather than non-syndromic CHD [6,7]. Therefore, it is not surprising that known CHD genes were enriched in this cohort. To identify non-syndromic causes of CHD, it is important to take into account previous findings, which have shown a significant excess of apparently deleterious inherited PTVs in unaffected parents [6]. To help address the challenges of identifying non-syndromic CHD genes, we have used an integrative approach, which has allowed us to look for novel CHD associations in a larger sample size, in a binary fashion. This will help to facilitate future studies.

Variable expression and reduced penetrance are common features in CHD, including in even well-established conditions such as Noonan Syndrome [49]. Most cases of CHD occur as a one off in the family and when recurrences do happen, CHD subtypes are more likely to be discordant than concordant [50]. This suggests that other modifying factors may be at play, including genetic, or environmental factors, or both. This presents a further challenge in identifying genetic causes of CHD. Moving forward, detailed phenotypic descriptions using a standardized system, and better data sharing strategies (including primary data), will facilitate further gene discovery and improved genotype-phenotype correlation in CHD subgroups.

In summary, we have performed an integrative analysis of CNVs, DNVs, o/e LOF ratio and expression during heart development amongst more than 10,000 CHD patients. Our analyses identify seven potential disease genes and mechanisms with novel association with CHD and strengthen previously reported associations.

## Materials and methods

### Ethics statement

This project was based on unidentifiable data and did not require approval by science ethics committees in Denmark or Christian-Albrechts-Universität in Kiel.

### Cohort description

Our cohort contains 7,958 CHD cases and 14,082 controls (see summary at **S1 Table**). Data from both affected and unaffected individuals were collected from 190 different CNV studies (**S2 Table**). Most of the CNV data included in the present study were assembled from public repositories, data available from literature as well as clinical data (see **S2 and S3 Tables** for a more detailed description). We sampled all available and accessible studies as of February 2018 cited in PubMed. We have focused on studies incorporating Caucasian samples (the larger sampled population) to decrease population effects. Given no primary data were available for most of the studies, we cannot exclude the possibility that non-Caucasian samples were included. Both non-syndromic and syndromic CHD cases were included and CNVs were mapped to the human genome build NCBI37/hg19. Phenotype information was reviewed and if possible standardised across studies, to ensure consistency and accuracy. We used as controls re-analysed samples from the Wellcome Trust Case Control Consortium 2, the Genetic Association Network (GAIN) and the Ottawa Heart Institute (**S3 Table**). In addition, we built a dataset from the two largest DNV studies in CHD published thus far, which include a total of 2,489 parent-offspring trios [6,7].

### Determining the gene sets

The compiled highly confident CHD gene list consisted of genes with plausible disease-causing mutations in an interpretable functional region of a gene reported in association with CHD in

three or more unrelated individuals. Genes, where fewer than three reports of CHD exist, were also considered if there was available solid functional evidence, such as a mouse model that displays CHD or *in silico* evidence. CHD genes were then coded as either non-syndromic (isolated CHD), or syndromic based on published phenotypes. The list of genes associated with developmental disorders was derived from the Developmental Disorder Genotype to Phenotype (DDG2P) list maintained by Decipher and the European bioinformatics Institute [51]. All genes were annotated with either monoallelic or biallelic as appropriate, based on the published literature (**S13 Table**).

## CNV analysis

Only autosomal CNVs were included in the analysis. All the CNV boundaries were determined using genome build NCBI37/hg19. For the CNVs provided in hg18, we used the Assembly Converter (https://www.ensembl.org/Homo_sapiens/Tools/AssemblyConverter) build on CrossMap (http://crossmap.sourceforge.net/) to convert samples to NCBI37/hg19. Also, an extra validation step of all CNV boundaries was performed using the R-package *BSgenome.Hsapiens.UCSC.hg19*. Smaller and longer CNVs were filtered out by applying a size cutoff of 5 Kb and 20 Mb as lower and upper limit, respectively. It has been demonstrated before that smaller and larger CNV calls tend to have a high rate of false positives [52,53]. We removed CNVs overlapping more than 50% of telomeres, centromeres and segmental duplication regions. In addition, we computed the internal CNV frequencies by counting the number of relative overlaps (>50% reciprocal overlap) on the CNVs control subset divided by the total number of controls. The internal MAF was computed for deletions and duplications subsets separately. Only CNVs with a minor allele frequency (MAF) < 0.01 in controls and overlapping ten or more CNV platform calling probes (Affymetrix Array 6.0 and Illumina Human660W-Quad) were considered for downstream analysis. Our analysis was focused only on CNV deletions. The distributions of the number of CNV deletions per individual within the case and control groups were compared (two-sided Wilcoxon rank sum test) to evaluate the impact of the quality control filtering step (**S5 Fig**). After filtering, 6,746 cases (3, 929 harbouring CNV deletions) and 14,024 controls (12,585 harbouring CNV deletions) remained for further analysis. A region-based permutation test (using PLINK version 1.07, test '*—cnv-test-region—mperm 10000*') was used on the filtered set to perform a case-control association analysis. For the gene-based permutation analysis, we reported both the 'point-wise' empirical p-value (EMP1) and the empirical adjusted p-value (EMP2), which controls the family-wise error rate (FWER) (http://zzz.bwh.harvard.edu/plink/perm.shtml). In addition to the gene-centered permutation testing, a similar region-based permutation analysis was performed to access enrichment in known CNV deletion syndromes. All CNVs deletions passing QC filtering overlapping these regions were considered in the analysis. The region genomic coordinates and syndrome descriptions were downloaded from the *Database of genomic variation and phenotype in humans using Ensembl resources* (Decipher, https://decipher.sanger.ac.uk/disorders#syndromes/overview).

## CNV burden test on gene sets

A logistic regression-based burden test ('cnv-enrichment-test' in PLINK v1.7) [11] was performed on different gene sets (known CHD genes (non-syndromic/syndromic/biallelic/monoallelic), developmental disorder genes, known haploinsufficient disease genes [54], autosomal recessive disease genes [55,56] and low observed/expected LOF ratio genes, **S13 Table**) using the rare CNV deletions passing the quality control and filtering stage. For every gene set examined, the binary phenotype (CHD case or control) was regressed on the number of genes

disrupted by one or more CNVs. The averaged CNV size and the number of segments per individual were used as covariates into the model to control for potential differences between cases and controls as suggested by Raychaudhuri *et al* [11]. In addition, the PLINK implementation of this test was slightly modified by including a third (categorical) covariate, the sample study ID, since we have assembled the CNV data from different sources.

## DNV analysis

The assembled DNV dataset (Sheet A in **S14 Table**) was re-annotated using the Variant Effect Predictor (VEP version 90) tool. All the DNVs included in this study were validated with the VariantValidator tool [57] (Sheet B in **S14 Table**). Based on the VEP annotation, we classified every variation into three major functional groups as follows: a) Protein truncation variant (stop_gained, splice_acceptor, splice_donor, frameshift, initiator_codon, start_lost, conserved_exon_terminus), b) missense variant (stop_lost, missense, inframe_deletion, inframe_insertion, coding_sequence, protein_altering) and c) silent variant (synonymous). Variants with minor allelic frequency (MAF) > 0.01 in gnomAD database were excluded from the analysis. The rates of rare DNVs (MAF < 0.01) in both DNV studies [6,7] were compared (Poisson test) for different variant consequence groups (PTV, missense and synonymous). No significant differences were found between the DNV rates for any of the evaluated groups (**S9 Table**). *De novo* variation recurrence significance testing was performed to evaluate the impact of DNVs at gene level using the Mupit tool [12]. By default, Mupit uses the sequence-specific variation rate published by Samocha *et al* [58]. A second test, DeNovoWEST [13], was used to assess gene-wise *de novo* variation enrichment. DeNovoWEST assigns a variation severity score (based on the variant consequences and the CADD score) to all classes of variants as a proxy of its deleteriousness. For each tested gene, the minimal *p-value* obtained from Mupit and DeNovoWEST was reported ($P_{dnv}$). The corrected $P$ value was computed using the Bonferroni method with n = 18,272.

## Inferring differentially and highly expressed genes

Differentially expressed genes (DEGs) were identified by comparing the gene expression profile in heart to kidney and liver at matched time points. We used maSigPro R-package [59] for inferring genes with dynamic temporal profiles from time-course transcriptomic data as previously described by Cardoso-Moreira *et al* [17]. As the input for maSigPro, we used the count per million matrix (CPM, output from EdgeR package) hosted in ArrayExpress (E-MTAB-6814). Genes which did not reach a CPM > 0.5 in at least five samples were excluded from the analysis. We ran maSigPro on the time scale measured in days post-conception using defaults parameters and only included time points with at least two biological replicates. A gene was selected as DEG if the $R^2$ (goodness-of-fit) parameter was higher than 0.50 and Bonferroni corrected $P < 0.01$. The $R^2$ parameter distinguish genes with clear expression trends from genes with 'flat' expression profile. **S12 Table** lists the final DEGs identified in the heart ($R^2 > 0.50$). To assess the gene expression levels in the heart, the RPKM matrix was used. Gene expression levels were averaged among samples in the different development stages of the heart as follow: early development (4wpc-8wpc), maturation (9wpc-20wpc), infant/adult (newborn-adulthood). Genes were ranked based on the computed mean expression.

## Identification of CNV/DNV enriched PPI sub-clusters

A protein-protein interaction network was constructed using the GeNets framework [20] and the information from InWeb [60] and ConsensusPathDB [61] protein-protein interaction databases. Nodes in the network correspond to proteins whereas edges represent their physical

interactions. The network was strictly seeded with 492 candidate genes, those with a significant adjusted *metaP* < 0.05 (Benjamini-Hochberg's false discovery rate, FDR). The PPI network was partitioned into overlapping sub-clusters using the in-built clustering method described in GeNets [20]. Only statistically significant sub-clusters (p-value < 0.05, permutation test) with at least 5 proteins were considered for further analysis. Finally, Gene Ontology enrichment analysis (Biological Process database 2018) of each identified sub-cluster was performed using the enrichr tool (https://maayanlab.cloud/Enrichr/).

## Supporting information

**S1 Fig. Distribution of overlapping CNVs, size and genes in 63 CHD loci. A) CNVs per gene.** Overlapping CNVs for each of the 528 significant candidate genes are shown as box-and-whiskers plots. Statistically significant difference was observed between the two distributions (Mann-Whitney test, ***: P<0.001). B) Size of loci in kilobase-pairs (kbp). C) Number of genes per locus. Median values are shown above each box.
(TIF)

**S2 Fig. Statistical framework to discover novel candidate CHD genes by integrating DNV and CNV deletions.** The workflow follows four major steps: 1) Data aggregation and quality control of both DNV data and CNV data, 2) DNV rate-based enrichment testing and CNV deletions case/control association analysis at gene level are performed independently, 3) the results are combined using the Fisher method and 4) P-values are Bonferroni corrected using the Independent Hypothesis Weighting method (IHW). As independent covariate for the IHW method, the o/e LOF ratio upper bound fraction (LOEUF) was used.
(TIF)

**S3 Fig. Heart expression pattern of the 21 significant genes at different heart development stages.** Panels show three different heart development stages: early development (red), maturation (green) and infant/adult (blue). The x-axis denotes the percentile rank of heart expression in the heart. The y-axis denotes the o/e LOF ratio upper bound fraction (LOEUF) from gnomAD. Dashed lines denote the threshold for highly expressed genes (expression rank > = 0.75) and highly LOF constrained genes (LOEUF < = 0.30).
(TIF)

**S4 Fig. The functional network enriched for proteins encoded by genes affected by CNVs and/or DNVs associated with CHD.** Ten sub-clusters were identified using GeNets. Proteins are shown as nodes, interactions as edges. Enrichment for CNVs (blue), DNVs (green) or both independently (purple) are highlighted. Proteins with no specific enrichment for CNV and/or DNVs but with B-H adjusted *metaP* < 0.05 are highlighted in red. The size of the circles denotes if the gene was found significantly highly and/or differentially expressed in the heart (large circles: significant expression; small circles: non-significant).
(TIF)

**S5 Fig.** Distribution of the number of CNV deletions per individual in both control and CHD case cohorts before (A) and after (B) applying the quality control filtering approach. Differences between the distributions were tested using a two-sided Wilcoxon rank sum test. ****: P<0.0001.
(TIF)

**S1 Table. Number of probands in the CNV cohort.** Stratified by CHD cases/controls and CNV type (deletion/duplication).
(XLSX)

**S2 Table. Sources of the CNV-case cohorts used in this study.**
(XLSX)

**S3 Table. Sources of the CNV-control cohorts used in this study.**
(XLSX)

**S4 Table. Gene set-based logistical regression enrichment CNV analysis.**
(XLSX)

**S5 Table. Gene set-based logistical regression enrichment CNV analysis stratified by observed/expected LOF ratio (from gnomAD).**
(XLSX)

**S6 Table.** Sheet A) Gene-based case/control CNV-deletions permutation testing (PLINK results). Sheet B) Significant locus (Locus ID and contributing genes).
(XLSX)

**S7 Table. Case/control CNV permutation testing on known deletion syndrome (PLINK results).**
(XLSX)

**S8 Table. CNV case/control permuatation testing output from PLINK.** The table shows the 528 significant genes combining both CHD and TAA cases (CHD+TAA), the contribution of only CHD cases (only CHD) and the contribution of only TAA cases (only TAA).
(XLSX)

**S9 Table. Comparision of the DNV rates, stratified by variant consequences, between two independent cohorts.**
(XLSX)

**S10 Table. Gene-based DNV analysis.**
(XLSX)

**S11 Table. Meta-analysis of CNV/DNV stratified by CHD sub-types (Conotruncal and LVOTO).** Table shows the four genes with Bonferroni corrected metaP < 0.05. Cases/Controls: Number of cases and controls carrying CNV deletions overlapping the gene in the CNV analysis. $p\_cnv$: p-value from the CNV permutation test. nsDNV: Number of constrained non-synonymous variations in the *de novo* analysis. $p\_dnv$: p-value from the DNV analysis. *metaP*: combined p-value ($P_{dnv}$ and $P_{cnv}$) using the Fisher method. adj meta$P$: Bonferroni corrected p-value using independent hypothesis weighting (IHW) and LOEUF metric from gnomad as covariate.
(XLSX)

**S12 Table. Differentially expressed genes in the heart compared to kidney and liver.**
(XLSX)

**S13 Table. List of gene sets used in the CNV enrichment analysis.**
(XLSX)

**S14 Table.** Sheet A) List of the *de novo* variants analized in this study. Sheet B) Validation results of the DNVs from the VarinatValidator tool.
(XLSX)

**S15 Table. Rare CNV deletions (MAF 0.01) used in this study.** CNVs bounderies are determinated in genome build hg19.
(XLSX)

## Acknowledgments

We express our gratitude to the patients and their families for their participation in the analysed studies. We would like to thank the Genetic Association Information Network (GAIN) and dbGAP for making the data available. We would like to thank the Wellcome Trust Case Control Consortium (WTCCC) for making the data accessible https://www.wtccc.org.uk/info/access_to_data_samples.html. We used data from the the Deciphering Developmental Disorders (DDD) study. The DDD study presents independent research commissioned by the Health Innovation Challenge Fund, a parallel funding partnership between the Wellcome Trust and the UK Department of Health, and the Wellcome Trust Sanger Institute. The views expressed in this publication are those of the author(s) and not necessarily those of the Wellcome Trust or the UK Department of Health. We would like to thank the Pediatric Cardiac Genomics Consortium (PCGC) and dbGAP for making the data publicly available. This study makes use of data generated by the DECIPHER community. A full list of centres who contributed to the generation of the data is available from http://decipher.sanger.ac.uk and via email from decipher@sanger.ac.uk. We thanks to Margarida C. Moreira and the Kaessmann Lab by making accessible the human RNA-Seq data and the support for the data analysis. We thanks Rasmus Wernersson and Federico de Masi for facilitating the use of the protein-protein interaction database, InWeb, in this work. We thanks the Lage Lab and Taibo Li for their support with GeNets. And finally, we would like to thank all data submitters and collaborators for their contributions.

## Author Contributions

**Conceptualization:** Enrique Audain, Matthew E. Hurles, Bernard Thienpont, Lars Allan Larsen, Marc-Phillip Hitz.

**Data curation:** Enrique Audain, Anna Wilsdon, Jeroen Breckpot, Jose M. G. Izarzugaza, Anne-Karin Kahlert, Hashim Abdul-Khaliq, Mads Bak, Anne S. Bassett, Woodrow D. Benson, Felix Berger, Ingo Daehnert, Koenraad Devriendt, Sven Dittrich, Piers EF Daubeney, Vidu Garg, Karl Hackmann, Kirstin Hoff, Philipp Hofmann, Gregor Dombrowsky, Thomas Pickardt, Bernard D. Keavney, Sabine Klaassen, Christian R. Marshall, Dianna M. Milewicz, Scott Lemaire, Joseph S. Coselli, Michael E. Mitchell, Aoy Tomita-Mitchell, Siddharth K. Prakash, Karl Stamm, Alexandre F. R. Stewart, Candice K. Silversides, Reiner Siebert, Brigitte Stiller, Jill A. Rosenfeld, Inga Vater, Alex V. Postma, Almuth Caliebe, Lars Allan Larsen, Marc-Phillip Hitz.

**Formal analysis:** Enrique Audain, Jose M. G. Izarzugaza, Alejandro Sifrim, Lars Allan Larsen, Marc-Phillip Hitz.

**Funding acquisition:** Hans-Heiner Kramer, Matthew E. Hurles.

**Investigation:** Anna Wilsdon, Jeroen Breckpot.

**Methodology:** Tomas W. Fitzgerald, Alejandro Sifrim, Florian Wünnemann, Yasset Perez-Riverol, Gregor Andelfinger.

**Resources:** Thomas Pickardt, Ulrike Bauer, Hans-Heiner Kramer.

**Writing – original draft:** Enrique Audain, Anna Wilsdon, Jill A. Rosenfeld.

**Writing – review & editing:** Enrique Audain, Jeroen Breckpot, Tomas W. Fitzgerald, Anne-Karin Kahlert, Florian Wünnemann, Yasset Perez-Riverol, Hashim Abdul-Khaliq, Mads Bak, Anne S. Bassett, Woodrow D. Benson, Felix Berger, Ingo Daehnert, Koenraad Devriendt, Vidu Garg, Karl Hackmann, Kirstin Hoff, Philipp Hofmann, Gregor Dombrowsky, Bernard D. Keavney, Sabine Klaassen, Hans-Heiner Kramer, Christian R. Marshall, Dianna M. Milewicz, Scott Lemaire, Joseph S. Coselli, Michael E. Mitchell, Aoy Tomita-Mitchell, Siddharth K. Prakash, Karl Stamm, Alexandre F. R. Stewart, Candice K. Silversides, Reiner Siebert, Brigitte Stiller, Inga Vater, Alex V. Postma, Almuth Caliebe, J. David Brook, Gregor Andelfinger, Matthew E. Hurles, Bernard Thienpont, Lars Allan Larsen, Marc-Phillip Hitz.

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
