## [Decision Letter · Decision Letter 0]

4 Feb 2021

Dear Dr Hitz,

Thank you very much for submitting your Research Article entitled 'Integrative analysis of genomic variants reveals new associations of candidate haploinsufficient genes with congenital heart disease' to PLOS Genetics.

The manuscript was fully evaluated at the editorial level and by independent peer reviewers. The reviewers appreciated the attention to an important problem, but raised some substantial concerns about the current manuscript. Based on the reviews, we will not be able to accept this version of the manuscript, but we would be willing to review a much-revised version. We cannot, of course, promise publication at that time.

If you decide to revise the manuscript for further consideration at PLOS Genetics, please aim to resubmit within the next 60 days, unless it will take extra time to address the concerns of the reviewers, in which case we would appreciate an expected resubmission date by email to plosgenetics@plos.org.

[LINK]

We are sorry that we cannot be more positive about your manuscript at this stage. Please do not hesitate to contact us if you have any concerns or questions.

Yours sincerely,

Anthony B Firulli

Associate Editor

PLOS Genetics

Gregory Barsh

Editor-in-Chief

PLOS Genetics

Reviewer's Responses to Questions

**Comments to the Authors:**

Reviewer #1: This paper by Audain et al is a massive study collecting CNVs/DNVs data across multiple sources in an attempt to increase the power to determine involvement in CHD, and then explore these CNVs and DNVs to find individual genes and networks. They focused only on solidly pathogenic changes due to the nature of the case data, a trade off to using all CNVs that could identify new CNVs worth consideration as novel pathogenic variants. A large control group was selected across several datasets. The authors have used orthogonal analyses for several parts of the study (such as using Mupit and DeNovoWEST) when no one gold standard exists, and then considered the genes in the CNVs in different ways (straight association, as part of heart tissue expression, and then as differential expression, and as a network analysis). Overall, the study is quite well executed, and the paper provides nice confirmation of known information (specific CNVs, known genes and involvement of specific pathways). In addition, they identify several novel genes that warrant investigation.

The methods are scant in the description of the study design. The authors should state their search strategy for the literature and the public datasets chosen. Of the studies identified, how were they selected for inclusion or exclusion? Were the CNV boundaries determined using the same build of the genome? How did the authors determine the controls and cases were of similar ancestral groups? While this is likely less of a problem for CNVs, it would be helpful to note any differences between groups. Did the authors include all genes within a CNV, or was there some selection?

A definition of the different gene sets used in the analysis is needed. For example, it is not clear what difference there is between monoallelic and non-syndromic sets.

Was the phenotype information clean enough and were there enough individuals to explore the CHDs by subgroups? The genetic architecture is quite different for instance between conotruncal defects and left sided defects.

The authors should also acknowledge an additional bias of the case data collected. The CNVs are highly selected from only the cases published due the presence of a CNV so are not representative of all individuals with CHD.

Reviewer #2: In this manuscript, the authors present their study on an integrative analysis to identify copy number variants (CNVs) and de novo intragenic variants (DNVs) in patients with congenital heart disease (CHD) with sporadic thoracic aortic aneurysm. The authors used published and publicly available datasets of CNVs and DNVs identified in probands with CHD (10,447) to perform a genome-wide integrative meta-analysis. To achieve their goal the authors implemented a statistical approach which allows the integration of different types of genomic variants. Based on their study they identified 21 significant haploinsufficient CHD genes. Among these genes 14 are already known as CHD genes and 7 have not previously been associated with CHD. The authors evaluated expression pattern of these genes in the heart using available data from the recent human atlas study. In summary, this study used astutely available datasets to identify novel CHD genes.

Although the basic insight regarding the identification of novel genes involved in CHD is very interesting, there are some weaknesses in the data reported in this study.

Major concerns:

- My main concern is related to diversity of CHD used in this study. The authors used CHD as a generic denomination. There is no detail regarding the type and class of CHD used in this study. For instance, conotruncal and non-conotruncal groups would be a good indicator to better classify these CHDs in order to improve the analysis.

- I wonder if there is a bias in using these publicly available datasets since their integrative analysis revealed significant enrichment for the known CHD genes, which is mainly explained by the contribution of monoallelic CHD genes and syndromic CHD gene set. This should be better discussed and the authors should propose alternative analysis to identify non-syndromic CHD genes.

- It is very surprising to see that one of the most frequent syndromes (22q11/DiGeorge syndrome) is not among the CNV deletion syndrome regions identified in this study. Particularly, the authors mentioned that the distribution of CNVs that overlap known microdeletion syndrome such as DiGeorge syndrome is overrepresented in the dataset. I wonder about the quality of the analysis and the filters applied to identify CHD genes.

- The authors used bulk-RNAseq dataset to determine if the 21 significant genes are expressed in the heart at different developmental time points. However, there is no information regarding the region of the heart where these genes are expressed. This information could be useful in relation to the type of CHD they are associated with.

Minor points:

- S6 Table: It is not evident to find the 528 genes. The authors should better highlight them).

- The authors should better explain why it is important to known that genes remain highly expressed in adulthood.

- It is not clear why the authors have conducted a systems-level analysis to identify global mechanisms if there is no mechanism proposed in this study.

Reviewer #3: The manuscript by Audain et al. performs a meta-analysis of copy number variants and de novo variants identified by exome sequencing from previously published studies. The case and control cohort sizes are large and the authors have established an extensive cohort. Using burden analyses the authors identified 21 genes associated with CHD including 14 previously known genes and 7 “novel” genes. In many ways this manuscript provides proof of principle that meta-analyses from a large number of studies, including single case studies, can be performed and platform results can be harmonized for both DNV and CNVs and this is an important contribution. However, the novel findings are limited and the goal of using integrative analyses to identify CHD disease genes in multifactorial inheritance is not accomplished here but it does provide a path forward. It would improve the impact of the manuscript to apply the expression and/or network analyses to a somewhat broader group of candidate genes.

Major comments:

1. The study focuses on CNV deletions but it would be useful to perform analyses for duplications as well particularly for overlaps with the deletions as this would further support dosage sensitivity and mechanisms.

2. The control population is not described well. How carefully phenotyped are these controls?

3. Please clarify how genes were assigned to syndromic, non-syndromic, monoallelic, biallelic gene sets and other gene sets. What proportion of the genes are shared between lists and how is this accounted for in analyses?

4. Previous studies from some of the authors of this study have demonstrated enrichment of CNVs in specific subphenotypes of CHD. Additional information and analyses performed on subphenotypes would be useful to include as well as whether any previous findings can be expanded.

5. For TAA, previous studies from some of the authors indicate a higher burden of CNVs in familial TAA. What is the representation of familial vs sporadic in this meta-analysis and can any additional comments be made about CNV burden?

6. In Table 7 the method for identifying CNVs in controls needs explanation or better descriptors. Why are the CNV counts so high in well described genomic disorders such as 22q11.2/VCFS/DiGeorge (986 controls), 1p36 deletion syndrome (886 controls) or Cri du Chat (1488 controls). These values should also have established a MAF > .01 prompting filtering. More broadly, it is surprising that TBX1 was not identified in the burden analyses. Could the authors comment particularly as it was noted in the discussion that 22q11.2 cases were overrepresented and so TBX1 seemingly should have shown significant association by burden testing.

7. Some of the “novel” genes cause syndromes that do have well described CHD such as Wolf-Hirschhorn or ARID1B. The manuscript could be improved by a better discussion of modifiers or oligogenic mechanisms and reduced penetrance of CHD as the rule even in genetic syndromic conditions.

8. The data on TAA do not contribute significantly to the manuscript.

9. The manuscript is difficult to follow from an analysis standpoint at times. It would benefit from a flow diagram of analyses and/or including more of the methods within the manuscript.

Minor:

1. The size cutoff of 5 kb and 20 Mb is relevant and should be moved to the methods of the main manuscript.

2. Fig. S4 – what does purple denote?

3. Fig. 5 - the x axis (Score) needs an explanation in the legend

4. Heading p. 12 CNV/DNVs hinder the function of specific protein complexes should be reworded since this has not been proven by this study.

**Have all data underlying the figures and results presented in the manuscript been provided?**

Reviewer #1: Yes

Reviewer #2: Yes

Reviewer #3: Yes

PLOS authors have the option to publish the peer review history of their article (what does this mean?). If published, this will include your full peer review and any attached files.

Reviewer #1: **Yes: **Kim L McBride

Reviewer #2: No

Reviewer #3: No

---

## [Decision Letter · Decision Letter 1]

24 May 2021

Dear Dr Hitz,

Thank you very much for submitting your Research Article entitled 'Integrative analysis of genomic variants reveals new associations of candidate haploinsufficient genes with congenital heart disease' to PLOS Genetics.

The manuscript was fully evaluated at the editorial level and by independent peer reviewers. The reviewers appreciated the attention to an important topic but identified some concerns that we ask you address in a revised manuscript

We therefore ask you to modify the manuscript according to the review recommendations. Your revisions should address the specific points made by each reviewer.

[LINK]

Yours sincerely,

Anthony B Firulli

Associate Editor

PLOS Genetics

Gregory Barsh

Editor-in-Chief

PLOS Genetics

Reviewer's Responses to Questions

**Comments to the Authors:**

Reviewer #2: The manuscript by Audain et al. has been well revised. The authors have completed significant revisions based on my comments. I find the current version of the manuscript considerably improved. The authors have replied to my main concern regarding the diagnosis of specific CHD type. They have performed a sub-analysis of the data based on specific groups of heart defects. They have also replied to my question concerning the possibly bias of their analysis since they used previously published datasets. The current version included a discussion on this specific point. The authors have satisfactorily answered my question regarding the 22q11 locus. Absence of detailed information about spatiotemporal expression of identified genes with CHD does not confirm the link these genes in the different part of the heart and the type of CHD in which they are involved. The authors should have performed a better analysis to reinforce this specific issue.

Reviewer #3: The authors have been responsive to the reviewers’ comments within the limitations of available data. The addition of the CHD phenotype subclassifications is useful and the discussion/interpretation of the findings is appropriate. The requested points for clarification have mostly been addressed. I have only minor comments.

I did not note in the original submission that CNVs were compiled for autosomes only. I would suggest the discussion/limitations make more explicit the fact that X-linked candidate genes will not be identified by the integrative approach.

The incorporation of the detailed methods is helpful. For the older platforms in the study, was the minimum CNV interval or maximum CNV interval (< 20 Mb) used? Are platform differences the explanation for the persistent differences between cases and controls even after filtering (FigS5)?

**Have all data underlying the figures and results presented in the manuscript been provided?**

Reviewer #2: Yes

Reviewer #3: Yes

PLOS authors have the option to publish the peer review history of their article (what does this mean?). If published, this will include your full peer review and any attached files.

Reviewer #2: **Yes: **Stéphane ZAFFRAN

Reviewer #3: No

---

## [Editor Report · Decision Letter 2]

23 Jun 2021

Dear Dr Hitz,

We are pleased to inform you that your manuscript entitled "Integrative analysis of genomic variants reveals new associations of candidate haploinsufficient genes with congenital heart disease" has been editorially accepted for publication in PLOS Genetics. Congratulations!

Yours sincerely,

Anthony B Firulli

Associate Editor

PLOS Genetics

Gregory Barsh

Editor-in-Chief

PLOS Genetics

Comments from the reviewers (if applicable):

**Data Deposition**

http://datadryad.org/submit?journalID=pgenetics&manu=PGENETICS-D-20-01841R2

**Press Queries**

---

## [Editor Report · Acceptance letter]

26 Jul 2021

PGENETICS-D-20-01841R2 

Integrative analysis of genomic variants reveals new associations of candidate haploinsufficient genes with congenital heart disease 

Dear Dr Hitz, 

We are pleased to inform you that your manuscript entitled "Integrative analysis of genomic variants reveals new associations of candidate haploinsufficient genes with congenital heart disease" has been formally accepted for publication in PLOS Genetics! Your manuscript is now with our production department and you will be notified of the publication date in due course.

With kind regards,

Agota Szep

PLOS Genetics

On behalf of:
